# Calcium Mobilization in Endothelial Cell Functions

**DOI:** 10.3390/ijms20184525

**Published:** 2019-09-12

**Authors:** Antonio Filippini, Antonella D’Amore, Alessio D’Alessio

**Affiliations:** 1Department of Anatomy, Histology, Forensic Medicine and Orthopedics, Unit of Histology and Medical Embryology, Sapienza University of Rome, 00161 Rome, Italy; antonio.filippini@uniroma1.it (A.F.); antonella.damore@uniroma1.it (A.D.); 2Istituto di Istologia ed Embriologia, Università Cattolica del Sacro Cuore, Fondazione Policlinico Universitario “Agostino Gemelli”, IRCCS, 00168 Rome, Italy

**Keywords:** angiogenesis, endothelial cells, endothelial dysfunction, calcium, NAADP, NO, second messengers

## Abstract

Endothelial cells (ECs) constitute the innermost layer that lines all blood vessels from the larger arteries and veins to the smallest capillaries, including the lymphatic vessels. Despite the histological classification of endothelium of a simple epithelium and its homogeneous morphological appearance throughout the vascular system, ECs, instead, are extremely heterogeneous both structurally and functionally. The different arrangement of cell junctions between ECs and the local organization of the basal membrane generate different type of endothelium with different permeability features and functions. Continuous, fenestrated and discontinuous endothelia are distributed based on the specific function carried out by the organs. It is thought that a large number ECs functions and their responses to extracellular cues depend on changes in intracellular concentrations of calcium ion ([Ca^2+^]_i_). The extremely complex calcium machinery includes plasma membrane bound channels as well as intracellular receptors distributed in distinct cytosolic compartments that act jointly to maintain a physiological [Ca^2+^]_i_, which is crucial for triggering many cellular mechanisms. Here, we first survey the overall notions related to intracellular Ca^2+^ mobilization and later highlight the involvement of this second messenger in crucial ECs functions with the aim at stimulating further investigation that link Ca^2+^ mobilization to ECs in health and disease.

## 1. Introduction

By a histological view, the endothelium is classified as a covering simple (monolayer) squamous epithelial tissue that lines the inner surface of the entire vascular tree from the heart to the smallest capillary, including the lymphatic vessels. Therefore, both vascular and lymphatic endothelial cells (ECs) are in direct contact with the lymph, the blood tissue and a variety of circulating cells. Although the vascular endothelium preserves its typical histological characteristics along most of the entire vascular system, ECs instead, display significant heterogeneity and differentiation mostly ascribable to the local environment found in different organs. Distinguishing morphological changes of ECs that display prominent cuboidal appearance are found, for instance, in high endothelial venules defined as specialized postcapillary venules found in lymphoid tissues that support high levels of lymphocyte migration from the blood [1]. In addition, the occurrence of a continuous, fenestrated and discontinuous endothelium, with distinct permeability features, is widely found throughout the body [2]. EC heterogeneity also takes account of distinct endocytic pathways, expression of specific intercellular junctions and the local composition of the endothelial surface layer [3,4,5], a 500-nm macromolecular coating of glycosaminoglycans of the apical surface of ECs that physically interacts with circulating blood cells. Despite their heterogeneity, ECs all share common functions, including the maintenance of homeostatic functions, the regulation of the blood flow, the inhibition of the coagulation cascade, the exchange of molecules between the blood and tissues and the modulation of the inflammatory response [6]. To accomplish these functions, ECs must operate as accurate sensors able to distinguish and handle a variety of extracellular cues in order to generate appropriate responses that guarantee the physiological condition in vivo [7]. Therefore, the precise differentiation of ECs and their functional status are fundamental for organ homeostasis and become crucial in pathological conditions affecting the cardiovascular system. Calcium ion (Ca^2+^) is a ubiquitous second messenger that contributes to regulate a variety of cellular processes in many electrically excitable and non-excitable cells including ECs [8]. The control of Ca^2+^ signaling in different EC types has engaged a great interest since a long time [9]. To this regard, these authors have discussed Ca^2+^ metabolism in ECs and, in addition, raised the crucial point about EC heterogeneity in the regulation of Ca^2+^ signaling that is still debated and currently under investigation. Many crucial vascular functions, such as the control of vascular tone and the secretion of vasoactive factors, are associated to specific fluctuation of intracellular calcium concentrations [Ca^2+^]_i_. As it occurs in many cell types most of the intracellular (Ca^2+^) in ECs is sequestered in the endoplasmic reticulum (ER) [9], while the role of mitochondria in calcium storage in this cell type is modest [10,11,12]. It has been estimated that in ECs about three quarters of the entire Ca^2+^ pool is sequestered into ER, while the remaining 25% constitutes the mitochondrial pool [10]. However, it is generally accepted that there is an overall cooperation between ER and mitochondria in the regulation of intracellular Ca^2+^ flux. Indeed, in many cell types, mitochondria typically function by sequestering cytosolic Ca^2+^ that is later recaptured by ER. Unfortunately, mitochondria are efficient makers of reactive oxygen species (ROS) such as the superoxide anion (O_2_^−^), hydrogen peroxide (H_2_O_2_) and hydroxyl radical (OH) all potentially dangerous for the host cell as well as involved in a multitudinous of physiological cellular functions [11,12,13]. Since ROS generate biochemically as by-products during mitochondrial electron transport, cells have developed a number of defense mechanisms aiming at preventing the oxidative stress. On the other hand, ROS also act as potent signalling agents that promote specific cellular mechanisms such as proliferation and differentiation [14,15,16]. Of note, a number of recent studies suggested that ROS production and Ca^2+^-dependent signals are two intimately integrated mechanisms that actively interplay. Indeed, it has been suggested that high concentration of mitochondrial Ca^2+^ stimulates ROS production while the regulation of Ca^2+^ signal can be redox-dependent [17,18,19]. Unfortunately, the exact molecular mechanisms that regulate the crosstalk between calcium and ROS signaling in ECs remain elusive. Furthermore, during the last decade, a number of intracellular acidic compartments have been proposed to serve as intracellular Ca^2+^ stores in many cell types [20]. These include a variety of endo-lysosome-like organelles equipped with the recent discovered Ca^2+^-permeable channels through which nicotinic acid-adenine dinucleotide phosphate (NAADP) mobilizes Ca^2+^ (see below). In addition, contributing to make the scenario even more complex there is the presence of a variety of intracellular heterogeneous calcium binding proteins (CBP) that specifically bind Ca^2+^ and finely regulate its cytosolic concentration [21]. Among these, some members of the S100 CBP family have called high interest in ECs for their contribution to regulate crucial mechanisms such as the control of cell cycle and senescence and more in general for their potential role in vascular functions [22,23,24].

### 1.1. Generation of Intracellular Second Messengers

The term second messenger refers to a variety of small and diffusible molecules that convey and amplify primary signals originating from the cell-surface receptors to intracellular effector proteins [25]. The major classes of second messengers released upon extracellular (primary) stimuli include cyclic nucleotides (such as cAMP and cGMP), inositol 1, 4, 5-triphosphate (IP3), diacylglycerol (DAG) and Ca^2+^ [25]. In addition, cell expressing ADP-ribosyl cyclases also synthetize cyclic ADP-ribose (cADPR) and NAADP from NAD and NADP, respectively [26]. Both cADPR and NAADP are two well-known Ca^2+^ mobilizing agent in many cell types, whose production involve distinct intracellular compartments [27,28]. More recent studies have reported the existence of a NAADP-synthesizing CD38 restricted to lysosomes/endolysosomal compartments of both human and mouse cells [29,30]. Ca^2+^ represents one of the most studied and ubiquitous second messengers that regulates a variety of intracellular events in many cell types, including ECs. The amount of cytosolic Ca^2+^, which is kept at a very low concentration (≤ 10^−7^ M) in resting cells, can either increase due to the influx from the extracellular space through specific calcium channels expressed on the cell surface or by release from intracellular stores such as ER (or sarcoplasmic reticulum in muscle cells). Ca^2+^ mobilization from intracellular stores typically occurs after the recruitment of either phospholipase C (PLC) β by G protein-coupled receptors (GPCRs) or PLC-γ isoforms by tyrosine kinase receptors and nonreceptor tyrosine kinases that induce the hydrolysis of the phospholipid phosphatidylinositol-4,5-bisphosphate (PIP_2_) to generate IP3 and DAG. While DAG remains close to the plasma membrane where it activates protein kinase C (PKC) [31,32], IP3 diffuses through the cytosol and binds to IP3 receptors (IP_3_Rs) expressed on ER membrane causing the increase in [Ca^2+^]_i_ [33]. Likewise, different types of Ca^2+^-selective intracellular channels termed ryanodine receptors (RyRs) trigger the release of Ca^2+^ from the sarcoplasmic reticulum (ER) in muscle cells [34]. Once released from intracellular stores, diffusible Ca^2+^ can either bind to several Ca^2+^-binding targets or it can recruit downstream effector proteins [35] that in turns drive a plethora of downstream mechanisms leading to contraction of muscle cells, cell adhesion, cell cycle progression, cell growth, cell motility, cell differentiation, fertilization and cell death. Moreover, although ER and its specialized form in muscle cells are by far considered the best characterized intracellular calcium store in both non-excitable and excitable cells, othercompartments displaying an acidic milieu have been reported to serve as a calcium depository in different cell types [20,36]. To date, among the most recent and potent calcium-mobilizing intracellular messengers from acidic compartments is the nicotinic acid-adenine dinucleotide phosphate (NAADP), whose function was first discovered in sea urchin eggs [37,38]. Ca^2+^ release from intracellular acidic compartments has been more recently linked to a novel class of membrane bound TPC expressed on lysosome-like organelles [39,40,41]. To date, three distinct TPCs termed TPC1, TPC2 and TPC3 (not present in humans [42]), have been identified within different types of intracellular acidic compartments like lysosomes and endosomes. Interestingly, the presence of functional NAADP-sensitive lysosome-like acidic compartments has been established in aortic ECs and it has been linked to nitric oxide (NO) synthesis and muscle relaxation [43]. In addition, in ECs, NAADP has been also found to contribute to crucial mechanisms such as histamine-induced release of von Willebrand factor [44] and VEGFR2-mediated signaling and neoangiogenesis [45,46], further supporting the role of TPCs as specific NAADP receptors expressed by lysosome-like organelles. Of note, it has been demonstrated that the flavonoid naringenin, by acting on human TPC2 channel activity, dampens NAADP-dependent intracellular Ca^2+^ responses to VEGF in ECs and impairs angiogenic activity in VEGF-containing matrigel plugs implanted in mice [47,48]. A recent study also linked TPC1 and NAADP-induced Ca^2+^ mobilization to the proliferation of metastatic colorectal cancer cells, suggesting that NAADP-induced signaling through TPC1 may represent an interesting mechanism to develop potential clinical targets in patients [49]. Whatever is the mechanism that led to the increase in cytosolic [Ca^2+^]_i_ and once specific downstream events have been activated, the amount of cytosolic Ca^2+^ rapidly decreases, bringing back the cell at its resting state. To this regard, both plasma membrane Ca^2+^ ATPases and sodium calcium exchangers came into play to extrude Ca^2+^ back into the extracellular space [50,51]. At the same time, the sarco/endoplasmic reticulum Ca^2+^-ATPase (SERCA) causes cytosolic Ca^2+^ storage into ER, re-establishing physiological cytosolic [Ca^2+^]_i_. Therefore, one may expect that any condition that interferes with the complex molecular machinery heading the series of event occurring during Ca^2+^ mobilization may affect normal behavior of ECs, leading to pathological conditions [52].

### 1.2. Calcium Machinery and Ca^2+^ Measurement

A multiplicity of vascular functions relies by the integrity of the endothelial layer lining the intima of blood vessels. Although ECs are non-excitable cells, changes in [Ca^2+^]_i_ which generate in response to disparate extracellular cues are crucial regulators of a number of vascular mechanisms linked to regulation of the vessel tone, inflammation, coagulation and regulation of vascular permeability, just to mention a few. A variety of ion channels expressed the cell surface of ECs mediate Ca^2+^ influx from the extracellular milieu that could be followed by intracellular Ca^2+^mobilization from internal stores. These include Voltage-Dependent Calcium Channels [53], Transient Receptor Potential Channels (TRP) [54,55], ORAI Family of Calcium Channels and Store-Operated Calcium Channels (SOC) [56,57,58]. The extensive functional and structural heterogeneity of the plasma membrane calcium channels has been largely reviewed [57,59,60]. In addition, some of these channels have been linked to vascular dysfunctions [61]. Although the amplitude and the duration of Ca^2+^ increase dependent on the type of stimulus and the vessel type, changes in [Ca^2+^]_i_ can be monitored in living cells. To measure in-cell changes in cytosolic [Ca^2+^]_i_ or agonist-induced Ca^2+^ mobilization, a number of fluorescent Ca^2+^ indicators are commercially available. These indicators have been first developed during the early 80′s and have soon became a fundamental tool for monitoring intracellular Ca^2+^ changes in many cell types [62,63,64]. Among these, ratiometric Ca^2+^ indicators such as Fura-2-acetoxymethyl ester (Fura-2 AM) is a widely used membrane-permeable indicator that once inside the cell is subjected to an enzymatic cleavage that removes the acetoxymethyl group allowing it to bind to free Ca^2+^. At this point, 340/380 nm excitation ratio of the molecules allows an accurate measurement of the intracellular Ca^2+^ concentration. Hovever, prevailing cell culture approaches may introduce modifications in morphology as well as expression and topography of ion channel of ECs in a way that could confound comparison to physiological conditions observed ex vivo or in vivo. Given the complexity of endothelial Ca^2+^ signaling, a major challenge is to clarify the spatial and temporal dynamics of Ca^2+^ mobilization in intact tissues [65,66]. To this end, the dynamic of the Ca^2+^ signals and its physiological implications have been investigated by a novel approach using intact endothelium from arteries [65,67,68,69]. Interestingly, to elucidate these dynamics events, Taylor and co-workersdeveloped a novel algorithmic process to evaluate basal Ca^2+^ signals within the endothelium of intact mouse mesenteric arteries [70].

### 1.3. Physiological Calcium Signaling in ECs

Despite its histological classification of an epithelial tissue and based on the numerous molecules such as vasoactive compounds, growth factors, cytokines and coagulation factors released by ECs, the endothelium is considered a genuine endocrine organ [71]. The endocrine feature of the endothelium adds to its well-known role of serving as a histological barrier separating the blood from the connective tissue underneath. This physical role turns into a more sophisticated as well as clinically relevant structure in specific areas of the body as it occurs in the blood–brain-barrier or the blood–thymus barrier [72,73,74]. However, taking into consideration the many functions played by the vascular endothelium, the capability of ECs to respond to extracellular stimuli such as neurotransmitters, growth factors and hormones is crucial in the regulation of the vascular tone. Notably, the secretion of a variety of vasoactive compounds, such as NO [75] and prostaglandin I_2_ (PGI_2_) by ECs relies on the capability of these cells to finely regulate [Ca^2+^]_i_ [76,77,78]. NO [75] is a soluble gas with a half-life of only few seconds synthesized by NO synthase (NOS) enzyme expressed in a variety of cell types, including red blood cells, platelets and ECs [79,80,81]. The activity of NOS enzymes depends on the presence of calmodulin [82], a cytosolic Ca^2+^-binding protein widely expressed in many eukaryotic cells. The talent of NO to induce Ca^2+^ mobilization from intracellular compartments, particularly ER, has been first described in ECs [83] and more recently it has been reported to involve ryanodine receptors [84]. Despite its short half-life, NO induces both short- and long-lasting effects in target cells by inducing at least three different waves of gene expression [85]. The well-known effect of NO generated by the endothelial NOS (eNOS) refers to its ability to relax vascular smooth muscle cells (VSMC) by increasing the activity of cytosolic cyclic guanosine monophosphate (cGMP) that induces Ca^2+^ uptake into intracellular stores and inhibits calcium–calmodulin myosin light chain kinase-complex formation. Indeed, although NO may count on different intracellular target, most of NO-dependent effects rely on the activation of the cGMP [86]. In addition, eNOS activity *per se* is regulated by a variety of proteins that act by either increasing or reducing its enzymatic activity [87]. Among proteins that behaves as a negative regulators of eNOS, there is caveolin-1 (cav-1) [88,89], the scaffolding protein of flask-shaped caveolae particularly enriched in the plasma membrane of ECs. Since caveolae have been involved in the regulation of intracellular Ca^2+^ levels in ECs and other cell types, it is not surprising that the perturbation of the caveolar network or the abnormal expression of cav-1 may results in the perturbation of eNOS activity and NO release from ECs. Considering the variety of vascular functions that depend by the bioavailability of NO, pathological conditions that affect eNOS activity may result in endothelial dysfunctions and cardiovascular diseases. These pathological conditions turn eNOS to a superoxide anion (O_2_^−^) producing enzyme instead of NO which contributes to cell injury and vascular diseases, a mechanism known as eNOS uncoupling [90]. Due to the reliance of eNOS from intracellular Ca^2+^ and the role that NO plays in blood pressure homeostasis, it is expected that several calcium channel blockers, or calcium antagonist, are employed to treat hypertension, although the precise mechanisms influenced by these drugs need more investigations.

Among the variety of agonists stimulating intracellular Ca^2+^ mobilization in ECs already mentioned above, it has been demonstrated that histamine through its H1 receptor and VEGF/VEGFR2 pathway play a relevant physiological role in the regulation of vWF release, capillary-like formation and in vivo angiogenesis, respectively [44,45]. Supporting these studies, Vinet et al., demonstrated that bradykinin and histamine induced calcium increase in ECs by means of different intracellular mechanisms based on the different sensitivity to thapsigargin. These authors claimed that bradykinin- and histamine-induced intracellular Ca^2+^ increase are of physiological relevance in modulating adrenal gland microcirculation [91]. The physiological role of a number of agonist-induced Ca^2+^ mobilization in ECs, such as angiotensin II, serotonin and acetylcholine has previously been extensively reviewed [8,92]. Their role has been demonstrated in activating phospholipase C-β1 followed by the generation of IP3 and DAG. On the other hand, growth factors such as platelet-derived growth factor and epidermal growth factor coupled to tyrosine kinase receptors activate phospholipase C-γ1 and elicit Ca^2+^ mobilization [8,92].

### 1.4. Role of Ca^2+^ in EC Permeability and Inflammatory Response

The endothelium serves as a crucial semipermeable barrier between blood and the interstitium. However, the inflammatory status induces changes in the permeability of this barrier, allowing access of plasma proteins to the surrounding tissues supporting vascular leaking. The regulation of vascular permeability is a hallmark of inflammation and is mediated by numerous inflammatory mediators such as histamine and thrombin. The ability of these molecules to induce intracellular Ca^2+^ mobilization has been widely demonstrated both in mast cells and ECs [93,94,95]. Nevertheless, VEGF, a major player able to activate the angiogenic pathway in ECs has been directly involved in the regulation of cell permeability via a Ca^2+^-dependent pathway [96]. In addition, we recently suggested that histamine-mediated release of von Willebrand factor (vWF) through H1 receptor from acidic stores requires the presence of functional NAADP receptors in ECs. Yet, under the same experimental condition, thrombin-induced vWF secretion was unaffected, demonstrating the obligatory role of NAADP exclusively in H1R-induced cell responses [44]. Due to the crucial role that vWF plays in the regulation of hemostatic functions, we proposed that targeting NAADP signaling in ECs may be beneficial to set in novel clinical strategies to counteract vascular diseases. ECs also play a crucial role in inflammation by responding to proinflammatory stimuli such as lipopolysaccharide (LPS), interleukin (IL)-1α and tumor necrosis factor (TNF). TNF exerts its function by binding to TNF receptor 1 and 2 that are differently distributed in ECs [97,98]. A soluble form of both receptors (sTNFR1 and sTNFR2) that are produced by shedding of the membrane bound molecules has been also described in ECs [99,100]. Although the functional significance of sTNFRs requires more investigation, sTNFR1 may function as a decoy receptor for TNF, thus reducing its proinflammatory functions [98]. Interestingly, while TNFR1 compartmentalization into lipid rafts/caveolae only partially affects TNF signaling in ECs [101,102], the presence of a functional caveolar network is critical for histamine-induced release of sTNFR1. By contrast, H1R-induced intracellular Ca^2+^ mobilization appears to be independent of the presence of functional lipid rafts [100], suggesting that the two mechanisms proceed independently each other. In Human Umbilical Vein Endothelial Cells (HUVEC), the release of s TNFR1 is strictly dependent on the expression of an integral membrane aminopeptidase, also known as ARTS-1 that associates with the calcium dependent protein nucleobindin-2, promoting TNFR1 release into the extracellular milieu via exosome vesicles [103]. Since the release of soluble TNFR1 is of primary importance to regulate TNF-induced signaling, these findings indicate that the precise regulation of [Ca^2+^]_i_ contributes to modulate the inflammatory response of ECs. In addition, more recent studies in humans suggested that EC functions are affected by elevated circulating calcium levels that, indeed, must be finely regulated in physiological conditions in vivo. Several studies indicate a direct correlation between blood calcium level and the onset of cardiovascular diseases and atherosclerosis [104,105]. ECs play a crucial role in the early step of atherosclerosis by inducing upregulation of adhesion molecules such as vascular cell adhesion molecule (VCAM-1) that triggers the recruitment of circulating monocytes [106,107] in response to inflammatory cytokines such as TNF [108]. Due to their specific position in the vascular system, ECs can serve as a sensor of extracellular Ca^2+^ level and any increase of the ion in the bloodstream will increase [Ca^2+^]_i_, which is fundamental to upregulate the expression of endothelial adhesion molecules [109,110]. These findings suggest a direct role of ECs in balancing both circulating and intracellular Ca^2+^ levels, which is crucial to regulate the inflammatory response and the onset of cardiovascular diseases.

### 1.5. Calcium Signaling in Normal and Pathological Angiogenesis

New blood vessel formation or angiogenesis not only represents a crucial mechanism in which ECs are the principal players, but plays a critical role during tumor growth and metastatic spread. Tumor cells obtain oxygen and nutrients by inducing the formation of new blood vessels from the local vasculature, giving rise to new vascular sprouting that can grow toward and infiltrate into the tumor mass sustaining its persistence and progression. As the tumor increases in size, nutrient deprivation and the resulting tumor hypoxia induce the release of soluble factors that "switch" ECs from a quiescent to an active status, an essential step that triggers new blood vessels formation [111,112]. A great variety of molecules have been identified that serve as angiogenic activators. These include VEGF, basic fibroblast growth factor (bFGF), angiogenin, transforming growth factor (TGF)-α, TGF-β, tumor necrosis factor (TNF)-α, platelet-derived endothelial growth factor, granulocyte colony-stimulating factor, placental growth factor, interleukin-8, hepatocyte growth factor and epidermal growth factor [113], whose release is also stimulated by the tumor microenvironment itself. Tumor-derived VEGF attracts and stimulates ECs to produce metalloproteinases (MMPs) that weaken the extracellular matrix facilitating the migration of ECs into the surrounding tissue. Under these circumstances ECs redeploy themselves, giving rise to hollow tubes that further evolve into a mature network of new blood vessels [114,115]. Finally, the intervention of pericytes and mural cells, i.e., vascular smooth muscle cells, contributes to the final stabilization of the newly formed blood vessels [116]. It has been demonstrated that the maintenance of Ca^2+^ signaling is necessary for both tube formation in vitro and angiogenesis in vivo [117]. Moreover, studies carried out by either overexpressing a dominant negative of TRPC6 on microvascular ECs or by its pharmacological inhibition in HUVEC, demonstrated the obligatory contribution of this channel to VEGF-mediated increase of intracellular Ca^2+^ as well as for the activation of VEGF-induced proliferation and angiogenesis [118,119]. In addition, cancer cells can mimic ECs in the formation of tubular-like structures, a phenomenon known as vasculogenic mimicry (VM) [120,121] observed in many cancers. To this regard, VE-cadherin, Notch and hypoxia-inducible factor 1-α (HIF1-α) are thought to be the most relevant signaling molecules involved in VM [122]. More recently, it has been reported that both intracellular and extracellular Ca^2+^ levels along with the involvement of ανβ3 and ανβ5 integrins play a crucial role during the development of capillary-like structures in melanoma [123]. Another study carried out on different cancer cell lines including human fibrosarcoma, breast adenocarcinoma and skin melanoma demonstrated the contribution of integrin β1 in the formation of VM-like structures. In particular, the administration of EGTA, a Ca^2+^ chelator, inhibited the formation of VM-like network, a phenomenon that was rescued by the addition of CaCl_2_ [124].

## 2. Conclusions

Undoubtedly the growing number of studies on Ca^2+^ signaling has greatly improved our knowledge about the control of fundamental mechanisms that occur in ECs. Although the crucial role of Ca^2+^ in VEGF-mediated signaling in ECs has been widely reported, understanding of the precise molecular mechanisms through which variation of the [Ca^2+^]_i_ controls the equilibrium between pro- and antiangiogenic factors remains crucial to the comprehension of the dysfunctions of the vascular endothelium. We believe that the use of genetic targeting of critical molecules involved in Ca^2+^signaling and the further characterization of the mechanisms controlling Ca^2+^ mobilization from intracellular stores in ECs is crucial to develop clinical strategies to counteract vascular dysfunctions that may lead to severe cardiovascular diseases. Moreover, although the effects of Ca^2+^ antagonists in ECs remain quite controversial, deepening the molecular mechanisms activated by these drugs could improve our knowledge of how Ca^2+^ signals originating from ECs contribute to ameliorate vascular dysfunctions, thereby reducing the risk of severe cardiovascular diseases such as atherosclerosis, hypertension, stroke and peripheral arterial disease. Notably, the expression of low voltage activated T-type Ca^2+^channels and their involvement in the reduction of vascular dysfunctions [53,125] raised great interest in the use of specific Ca^2+^antagonists in ECs. However, their specific activity within the vascular bed remain still unclear and need further investigation. Though we did not address this topic in this work, it has been also reported the higher sensitivity as well as an increased intracellular Ca^2+^ levels in endothelial progenitor cells compared to the adult ECs [126,127]. The discovery that endothelial colony forming cells isolated from different sources, i.e., peripheral vs. umbilical cord blood, exhibit different arrangement of Ca^2+^ channels and different angiogenic capabilities should encourage further investigation aiming at linking Ca^2+^ signaling to revascularization. In conclusion, the main purpose of this review is to focus the reader’s attention on the calcium machinery in EC functions and to stimulate further investigation that may contribute to develop novel clinical strategies helpful in the field of vascular diseases.

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
