# Peer review of "Calcium Mobilization in Endothelial Cell Functions"

_ijms, 2019, doi:10.3390/ijms20184525_

Round 1

Reviewer 1 Report

Analysis of the mechanisms of [Ca2+]i regulation in endothelial cells with relation to the physiological and pathophysiological processes is an actual task. In a review by Filippini et al. a large amount of information regarding this problem is presented. The review is well-structured. There are some comments. The overall notions related to intracellular Ca2+ mobilization are presented to some extent in a simplified form. To my opinion, this review could be improved by introducing more data directly related to endothelial cells. After reading of the summary, one would expect to receive an answer to the question what are the features of calcium regulation in ECs compared to other cell types and are there differences in calcium metabolism in different types of endothelium with different functions. With regards to this the data concerning integral calcium signaling of ECs in monolayer could be presented (Adams et al., FASEB J. 1989 Oct;3(12):2389-400). In section 1.2 data related to different calcium channels in EC presented in a too lapidary style. It is worthwhile to describe the roles of these channels in ECs. In 1.3 section it would be interesting to get more information about  the agonists and receptors acting on [Ca2+]i in ECs and to describe, if possible, how calcium signaling relates to their physiological effects. It might be worthwhile to combine 1.3 and 1.4. There is no information about pathological calcium signaling in ECs in 1.3 section while it is present in 1.5. I would suggest to discuss briefly the influence of reactive oxygen species on calcium signaling in the review.

Other comments.

Introduction.

Line 57-60. The assertion that mitochondria in ECs are mostly involved in cell signaling rather than serving as a calcium store needs to be explained and backed up with references.

Line 66-69. This sentence is not quite clear. Do the authors refer TPC, TRP and InsP3 channels to receptors located in acidic vesicles? TPCs not necessarily possess NAADP-binding site.

Generation of intracellular second messengers

In this section a general and well-known information is presented. However, some details are omitted. Phospholipase C gamma is not mentioned. I would suggest to include cADPR in the list of the main second messengers. What is known about the receptors coupled with cADPR and NAADP synthesis in ECs and other types of cells?

Line 92. Why for sarcoplasmic reticulum SE abbreviation is used?

Line 103. It should be mentioned that TPC3 is absent in humans.

Author Response

Author’s response (indicated in italic) to Reviewer #1

To my opinion, this review could be improved by introducing more data directly related to endothelial cells. After reading of the summary, one would expect to receive an answer to the question what are the features of calcium regulation in ECs compared to other cell types and are there differences in calcium metabolism in different types of endothelium with different functions. With regards to this the data concerning integral calcium signaling of ECs in monolayer could be presented (Adams et al., FASEB J. 1989 Oct;3(12):2389-400).

We thank the reviewer for her/his suggestions. We have accordingly reorganized and improved the introduction by citing the data presented by Adams et al. concerning the regulation of Ca2+ metabolism in ECs. In addition, we have also added new references concerning Ca2+-dependent mechanisms in ECs throughout the manuscript.

In section 1.2 data related to different calcium channels in EC presented in a too lapidary style. It is worthwhile to describe the roles of these channels in ECs

We agree with the reviewer that the description of the different type of calcium channels is lapidary. Our aim was, however, not to specifically focus on the functions and structures of the large variety of plasma membrane calcium channels but rather to discuss Ca2+-dependent mechanisms in EC. To this regard, we have included  a statement in the text (section 1.2) indicating three more citations that specifically review plasma membrane calcium channels in the vascular endothelium.

In 1.3 section it would be interesting to get more information about  the agonists and receptors acting on [Ca2+]i in ECs and to describe, if possible, how calcium signaling relates to their physiological effects.

As suggested, we expanded and improved the section 1.3 by introducing more information about  the agonists and receptors acting on [Ca2+]i in ECs, including the new appropriate references and reviews.

It might be worthwhile to combine 1.3 and 1.4.

As suggested by the reviewer,  we combined the sessions 1.3 and 1.4 and we have accordingly renumbered the other paragraphs.

There is no information about pathological calcium signaling in ECs in 1.3 section while it is present in 1.5.

We thank the reviewer for noticing the misprint present in a previous version of the ms with a different organization of the sections. Please note that now the section 1.3. was retitled to “Physiological calcium signaling in ECs.”

I would suggest to discuss briefly the influence of reactive oxygen species on calcium signaling in the review.

As suggested by the reviewer we have included few more lines about the crosstalk between reactive oxygen species and calcium signaling (from line 67 of the introduction).

Introduction.

Line 57-60. The assertion that mitochondria in ECs are mostly involved in cell signaling rather than serving as a calcium store needs to be explained and backed up with references.

We apologize with the reviewer for not being clear in elaborating this statement. We wanted to highlight the lower amount of Ca2+ stored into mitochondria with respect to ER. However, we have rephrased the indicated statement to make the concept more clear. Therefore, the previous statement at line 57-60, has been replaced with the following (line 61): “As it occurs in many cell types most of the intracellular (Ca2+) in ECs is typically sequestered in the endoplasmic reticulum (ER) [9], while the role of mitochondria in calcium storage in this cell type is modest [10-12].

Introduction.

Line 66-69. This sentence is not quite clear. Do the authors refer TPC, TRP and InsP3 channels to receptors located in acidic vesicles? TPCs not necessarily possess NAADP-binding site.

We agree with the reviewer that this sentence can generate confusion. To this regard, we replaced the indicated statement with the new one at line 80 that reads: "These include a variety of endo- lysosome-like organelles equipped with the recent discovered Ca2+-permeable channels through which nicotinic acid-adenine dinucleotide phosphate (NAADP) mobilizes Ca2+(see below).” This concept is further developed in the session 1.1 titled Generation of intracellular second messengers.

Generation of intracellular second messengers

In this section a general and well-known information is presented. However, some details are omitted. Phospholipase C gamma is not mentioned. I would suggest to include cADPR in the list of the main second messengers. What is known about the receptors coupled with cADPR and NAADP synthesis in ECs and other types of cells?

We thank the reviewer for her/his valuable comment. We have further expanded the information reported in the session "1.1. Generation of intracellular second messengers", as indicated by the blue text in the manuscript. In addition, new references have also been added in the text.

Evidence for the activity of an ADPR cyclase and for a role of cADPR and NAADP in Ca2+ signaling have been established in a large number of mammalian cells and tissues, including pancreatic islet cells, cardiac myocytes, trachea, intestinal muscle, lymphocytes, sympathetic neurons, salivary and lacrimal grand cells, hepatocytes, hypothalamus and posterior pituitary and peritubular smooth muscle cells; to our knoledge there are no incontrovertible  data on endothelial cells. We believe that information about cADPR and NAADP synthesis in other types of cells may confuse the influence and goes beyond the focus of this review.

Line 92. Why for sarcoplasmic reticulum SE abbreviation is used?

We apologize with the reviewer for the carelessness in writing this abbreviation. We have replaced "SE" with "ER" (now at line  111) as correctly addressed by the reviewer. In addition, we have added "smooth ER in muscle cells" at line 115 to better specify the role of smooth ER in calcium mobilization in excitable cells.

Line 103. It should be mentioned that TPC3 is absent in humans.

We thank the reviewer for noticing this inaccuracy. We have modified the indicated statement at line 109 and added the appropriate reference (new citation number 42).

Reviewer 2 Report

Fillipini et al provide a brief overview of some important aspects of calcium ion (Ca2+) as a mediator of endothelial regulation, focusing mainly on its role as a second messenger in controlling both physiological and pathological processes.

Significant concerns:

The authors discuss the measurement of intracellular Ca2+concentrations, but do not provide any comments acknowledging the latest developments in the field that focuses on the quantification of dynamic events in real time to determine spatial and temporal endothelial Ca2+signaling events occurring in intact tissues.

Due to the ubiquitous nature of Ca2+signaling, the authors should more clearly outline their vision as to how Ca2+channel blockers will be used clinically (beyond our current uses). These blockers have effects in several other cell types such as smooth muscle cells, and thus while targeting endothelial dependent Ca-mediated processes is logical, ensuring therapeutic specificity will likely be quite complex.

The conclusion section of this article focuses mainly on the previous section of the paper and does not truly encompass the many aspects of Ca2+ mobilization discussed in the entire manuscript.

Author Response

Author’s response (indicated in italic) to Reviewer #2

The authors discuss the measurement of intracellular Ca2+concentrations, but do not provide any comments acknowledging the latest developments in the field that focuses on the quantification of dynamic events in real time to determine spatial and temporal endothelial Ca2+signaling events occurring in intact tissues.

We agree with the reviewer that this topics deserves more informations. Therefore, we added a new part at the end of section 1.2.

Due to the ubiquitous nature of Ca2+signaling, the authors should more clearly outline their vision as to how Ca2+channel blockers will be used clinically (beyond our current uses). These blockers have effects in several other cell types such as smooth muscle cells, and thus while targeting endothelial dependent Ca-mediated processes is logical, ensuring therapeutic specificity will likely be quite complex.

The conclusion section of this article focuses mainly on the previous section of the paper and does not truly encompass the many aspects of Ca2+ mobilization discussed in the entire manuscript.

We absolutely agree with the reviewer about her/his concerns about the specificity of calcium antagonists in ECs. Althought it represents a complex as well as wide topic we thought it was important to mention these molecules in the work. However, we have extensively rephrased the conclusion section which also includes additional references.

We believe that the revised manuscript fully address the referee’s comments and is now acceptable for publication in the International Journal of Molecular Sciences.

Round 2

Reviewer 2 Report

No further comments at this time.